# An Audit and Survey of Informal Use of Instant Messaging for Dermatology in District Hospitals in KwaZulu-Natal, South Africa

**DOI:** 10.3390/ijerph19127462

**Published:** 2022-06-17

**Authors:** Christopher Morris, Richard E. Scott, Maurice Mars

**Affiliations:** 1Department of TeleHealth, School of Nursing & Public Health, College of Health Sciences, University of KwaZulu-Natal, Durban 4041, South Africa; ntc.ehealthconsulting@gmail.com (R.E.S.); mars@ukzn.ac.za (M.M.); 2Department of Community Health Sciences, Cumming School of Medicine, University of Calgary, Calgary, AB T2N 4N1, Canada; 3College of Nursing and Health Sciences, Flinders University, Adelaide 5042, Australia

**Keywords:** teledermatology, WhatsApp, instant messaging, legal, regulatory and ethical concerns, guidelines, HPCSA

## Abstract

Background. In KwaZulu-Natal (KZ-N), South Africa, recent reports have indicated that spontaneous use of smartphones has occurred, providing access to specialist dermatological care to remote areas. This informal use has raised a number of practical, legal, regulatory, and ethical concerns. Aim. To assess the nature and content of WhatsApp messages sent to dermatologists, to determine the referring doctors’ reasons for, and satisfaction with, their interactions, as well as their knowledge of legal, regulatory, and ethical requirements. Methods. A retrospective study of WhatsApp messages between referring doctors and dermatologists, as well as a cross-sectional survey of doctors working at district hospitals in KZ-N who used IM for teledermatology. Results. Use of IM (primarily WhatsApp) for teledermatology was almost universal, but often not considered ‘telemedicine’. Few referring doctors were aware of South Africa’s ethical guidelines and their requirements, and few of those who did followed them, e.g., the stipulated and onerous consent process and existing privacy and security legislations. No secure methods for record keeping or data storage of WhatsApp content were used. A desire to formalize the service existed. Conclusions. Based upon these findings, it was proposed that a number of described steps be followed in order to formalize the use of IM for teledermatology.

## 1. Introduction

eHealth, the use of information and communication technologies (ICT) for health [1], has long been identified as a potential means of improving access to equitable, affordable, and universal access to care, especially in the developing world. The World Health Assembly’s 2005 resolution 58.28 on eHealth called on member countries to develop eHealth strategies to facilitate the implementation and uptake of eHealth [2]. In response, many developing countries have drafted eHealth strategies, but implementation and sustained uptake of eHealth remains poor [3]. Telemedicine, the use of ICT to provide healthcare over distance [4], is a logical approach to overcoming shortages of, and problems of access to, healthcare professionals, especially medical specialists in the developing world. There have been many papers over the past two decades identifying and reporting the factors that need to be taken into account to ensure a successful telemedicine service [5,6,7], but there are still few sustained and scalable telemedicine endeavors in the developing world [3].

mHealth, the use of mobile devices, especially mobile phones, for eHealth is seen as a way of overcoming many issues (e.g., infrastructural costs, connectivity, bandwidth, expensive videoconferencing equipment, and reliable power), that have inhibited telemedicine in the developing world. As mobile phones have evolved from basic features to smartphones, so has the use of instant messaging (IM). What started on cellular networks as SMS (short message service; text messages limited to 160 characters) evolved to MMS (multimedia messaging service) and the widespread use of Internet-based IM applications such as WhatsApp, WeChat, and Messenger with the ability to acquire images, video, and audio and link them to text messages.

Various reviews have documented the evolution, since 2013 [8], of WhatsApp use in clinical medicine for both one-on-one consultation and one-to-many chat groups, usually within clinical departments [9,10]. While initially mostly reported in developing countries, recent papers have identified its growing use and acceptance in the developed world [11,12,13,14]. Few WhatsApp clinical activities were formally planned and implemented, with healthcare practitioners developing their own informal telemedicine networks and services [9,15], a growing practice during the COVID-19 pandemic. They began because of the ubiquity, ease of use (no training required), low cost, and the rapidity of information sharing provided by IM. The informality and lack of guidelines [16] has focused attention on legal, regulatory, and ethical issues related to WhatsApp use. These include consent, data security, patient confidentiality, privacy, standard and quality of care, record keeping, and the role of a prior doctor–patient relationship [16,17,18].

Dermatology, by its visual nature, lends itself to telemedicine, both store-and-forward (asynchronous) and videoconferenced (synchronous). A review of the literature in 2018 [19] identified three papers [15,20,21] reporting the use of WhatsApp in clinical dermatology. A subsequent PubMed search in January 2021 found that only a further nine papers had been published [22,23,24,25,26,27,28,29,30]. Despite few papers on the use of WhatsApp in dermatology, anecdotal reports suggest that the use of WhatsApp by clinicians is greater than suggested in the literature.

There have been several attempts to introduce teledermatology in South Africa with the first report of a short-lived store-and-forward teledermatology program reported in the Eastern Cape province in 2002 [31] and an ongoing videoconference-based teledermatology program in the province of KwaZulu-Natal (KZ-N), first reported in 2008 [32]. In 2013, there were anecdotal reports that physicians in rural hospitals in KZ-N were using WhatsApp for teledermatology and burn referrals [15,33]. The teledermatology service was not planned, there was no budget, no training had been required, and the KZ-N Department of Health (DoH) had no knowledge of its existence. In the absence of any formal regional telemedicine service, physicians had found a way to deliver a teledermatology service themselves [15]. The nature, efficacy, and extent of these informal interactions is not known. If the informal teledermatology activity is effective, a better understanding is required of what is currently occurring so that it may be formalized and expanded. 

The study had two aims. The first was to assess the nature and content of the information sent to dermatologists, and the second was to determine the referring doctors’ reasons for, and satisfaction with, their current interactions using IM, as well as their knowledge of the legal, regulatory, and ethical requirements of its use.

## 2. Methods

To address the first aim, an audit was performed to assess the nature and content of the information sent to dermatologists using IM. To characterize the current situation regarding use of IM for dermatology, a cross-sectional descriptive survey was performed, with convenience sampling of doctors from randomly selected district hospitals.

### 2.1. Audit of Informal WhatsApp Teledermatology

Doctors at rural hospitals in KZ-N have been using WhatsApp IM for at least seven years to send cases, texts, and photographs to the dermatologists at the province’s medical school. As the dermatologists had kept no formal records of these cases either electronically or on paper, the only data available for analysis were the cases that they had not deleted from their phones. This necessitated a retrospective analysis of the available data provided by dermatologists at the local medical school. The dermatologists sent any WhatsApp chats and related photographs remaining on their mobile phone that were received from referring doctors for audit. This was done using the email feature in the WhatsApp IM application to transfer data from their phones to a secure university server. Specific data were extracted (Table 1).

### 2.2. Survey of Doctors at District Hospitals

A cross-sectional survey was undertaken of doctors working at district hospitals in KZ-N who used IM for teledermatology. The province has 37 district hospitals in 11 districts. These hospitals do not have specialist physicians on site and refer dermatology cases to one of two tertiary hospitals. Two rural district hospitals were randomly selected from each of the 11 districts. As one district had only one rural district hospital, an additional rural hospital was randomly selected from one of the other districts. The hospital managers at the selected hospitals were contacted by telephone and asked to complete the survey and request their staff to complete the survey either online or by the supplied hardcopy. Reluctance of some managers to participate necessitated inclusion of three alternate sites.

### 2.3. Survey Instrument

As part of a larger study on telemedicine use in KZ-N, doctors who used IM for teledermatology were invited to complete a 43-item questionnaire. The survey was carried out in late 2019, before the COVID-19 pandemic, at 25 of the 37 district hospitals in KwaZulu-Natal. The questionnaire covered four domains: respondents’ demographics; their access to technology; IM use in dermatology; and legal and regulatory knowledge and practice of consent, data security measures, and guidelines for telemedicine practice. The questionnaire was developed by the authors and included dichotomous yes and no, multiple response, and Likert scale questions, and one open-ended question. For two Likert scale questions, respondents could add additional comments if they wished. Prior to its distribution, the questionnaire was pre-tested by five people experienced in telemedicine and medical informatics. Questions were modified where necessary to avoid ambiguity. The questionnaire could be completed online via Google Forms or a paper copy.

This study was undertaken at the request of the KZ-N Department of Health eHealth Steering Committee. Ethics approval was obtained from the University of KwaZulu-Natal and the KwaZulu-Natal Department of Health ethics committees for both the audit of the dermatologists’ cases and the survey of the doctors at the district hospitals. All respondents consented to participate.

## 3. Results

### Audit of WhatsApp Cases Sent to Dermatologists

Three dermatologists emailed cases from their phones for analysis. They all indicated that they had deleted some cases in order to free-up storage space on their phones or because the cases were no longer of interest, or that they had been deleted when they upgraded their phone and had not saved the case messages and images. 

A total of 322 unique cases using WhatsApp were reported between February 2015 and April 2019. These were from five district hospitals and were sent by 64 doctors. All submitted cases received a response from the dermatologists. The median response time for the first response by the dermatologist to the referring doctor was 7.5 min and for the dermatologist to resolve a case from receipt of the first message was 16 min (Table 2). Over three quarters of the cases (252, 78%) were replied to in under 30 min and 280 (87%) cases in under 60 min. Of the 322 cases, 22 (7%) were sent after 5 pm, with ten of these cases having a response time greater than the 75th percentile. There were twelve cases with a response time of more than four hours and five cases of more than fifteen hours (two overnight). On one occasion, the consultant asked the referring doctor to phone to discuss the case. Most cases (252, 78%) were diagnosed, resolved, and closed by the consultant in a single consultation (a discrete series of chat messages). The time to conclusion was less than 30 min in 204 cases (63%), or less than 60 min for 252 (78%) of cases. 

In 12 cases, the referring doctor only sent photographs to which the consultant responded with an immediate diagnosis. There was no information available as to whether the photographs were sent following a telephone consultation. In 16 other cases, no photographs were sent with the text messages, but there was either a description of the case from which the consultant could advise (8), or it was patient follow-up information (8). 

A total of 1034 messages (3.3 ± 3.0 per case) and 755 photographs (2.3 ± 1.5 per case) were sent by the referring doctors, and 830 responses (2.6 ± 2.2 per case) were sent by the dermatologists. No data were available on the time spent by the dermatologist from opening the message to replying. 

The messages were relatively short. The average number of words per message sent by the referring doctor and dermatologist were 11 ± 6.5 and 13 ± 2.8, respectively, and the average number of words per completed case were 36 ± 21.6 and 35 ± 7.3 words, respectively. The cost per case to the referring doctor (average 3.3 messages and 2.3 photographs) was less than R 0.032 (USD 0.0021) and to the dermatologist (average 2.6 messages), less than R 0.0001 (USD 0.00001) (1 USD = R 15) [34].

Twenty patients (6%) were requested to travel to Durban, 18 to attend the dermatology clinic and 2 to attend the oncology clinic and undergo an MRI scan. Biopsy or other special investigations, to be undertaken locally, were requested in 50 and 27 instances, respectively. It should be noted that the district hospitals do not have dermatopathology facilities with biopsies sent either to the anatomical pathology department of the medical school or local private practices. The most common diagnoses were eczema (27), lichen planus (19), tinea corporis (12), contact dermatitis (11), bullous impetigo (10) and papular urticaria (10), and for the 23 infants (under 12 months), bullous impetigo (5), eczema (4), and xanthogranuloma (3). In 76 cases (24%), the referring doctor provided a diagnosis, of which 32 (42%) were congruent with the dermatologist’s final diagnosis.

The dermatologists did not express dissatisfaction with the quality of any of the 755 photographs received. On four occasions (1%), additional photographs were requested (specific view or time series). In one instance, the dermatologist asked the patient to take and submit ‘selfies’ for a number of days. Of sixty-three photographs showing the patient’s face (8%), 20 were of infants, and only two had the patient’s eyes covered to maintain anonymity.

The patient’s age was provided in 187 cases (58%) and the mean age was 24 ± 21.1 y, in a range of (0–99 y). Infants made up 18% of these cases. Sex was indicated in only 112 cases (35%), among which 73 (65%) were women or girls. No patients were identified by name, case number, hospital number, or other unique identifier. A medical history and symptoms were included in the referral message in 168 cases (52%). Patient’s consent to the IM consultation was recorded only once. 

## 4. Teledermatology Survey of District Hospital Doctors

### 4.1. Demographics

Eighty-one responses were received from doctors at 20 district hospitals. All but one doctor used WhatsApp. Five used both WhatsApp and an alternative local IM application, Vula, for teledermatology, and only one used just Vula.

About half of respondents were women (43, 53%), of whom 37 (86%) were under the age of 40 y. Men were older, with ten (26%) older than 50 y including six (16%) who were over the age of 60. There were eight community service doctors, each working at a different district hospital. The frequency of referrals ranged from ad hoc (32 respondents) to approximately weekly (14), monthly (23), and quarterly (7).

### 4.2. Technical Issues, Response Times, and Satisfaction

Most respondents did not have difficulty taking photographs (71, 88%) or attaching them to messages (67, 83%). The majority also desired a response time of less than 30 min (68, 84%), and 41 (51%) reported receiving a response to their messages in under 30 min, with many responses received within an hour (58, 72%). 

IM met the needs of most of the respondents (75, 93%). Although six people reported that it did not meet their needs, they still rated their satisfaction as three on a five-point Likert scale. The median score for satisfaction was four, with 60 respondents (75%) either satisfied or very satisfied with IM teledermatology, and three dissatisfied or strongly dissatisfied, two of whom desired a more rapid response. Respondents were equally satisfied with IM for teledermatology, whether consults were sent to a KZ-N dermatologist or to another doctor.

Most respondents felt that patients were always or mostly satisfied with the WhatsApp IM process (68, 84%), and always or mostly satisfied with their photographs being taken (67, 83%). Over half of the doctors (42, 52%) using IM for teledermatology did not consider this to be telemedicine.

### 4.3. Consultation

Advice provided by dermatologists was more likely to be followed than that given by other colleagues (Table 3).

Doctors at three hospitals had never sent a case to a dermatologist but used WhatsApp to send consults to doctors at other hospitals. 

### 4.4. Open-Ended Questions

Eleven doctors responded to the open-ended questions. 

Three doctors reported their views on the benefits of WhatsApp:


*“the process makes it easier to start treatment for the patient while they are making a review date with the dermatologist.”*
Physician 25


*“I think it’s very helpful especially in the peripheries.”*
Physician 57


*“Speeds up patient care also saving money/resources as not all patients need to be transferred.”*
Physician 50

In addition, one doctor felt that consultation through teleconference would be useful:


*“Consultation through teleconference will be useful to cut on travelling and waiting time for patients in rural areas.”*
Physician 69

When sending messages, 96% of doctors used their own data bundle at their cost, with one suggesting that doctors need to be provided with a monthly allowance, 


*“HCPs need to be provided with a data allowance monthly.”*
Physician 25

The need for wireless connectivity was noted,


*“Need wireless connectivity at work to make it easy.”*
Physician 29


*“There’s a single desk computer at the MO office in my hospital, connectivity is average, many times no connection. Therefore, we used our data and devices a lot. It would be so nice to have wireless or Wi-Fi at the hospital. Which is not a new thing in many parts of the world.”*
Physician 75

As was the need to raise awareness.


*“I was not aware of an actual Teledermatology arrangement prior to this survey. I have previously called and discussed patients with our dermatology referral Centre and then used WhatsApp to send them pics of the lesions. I do find this useful and I feel that we could save a lot more time by having access to a designated teledermatology line as we would not have to find a dermatologist to discuss patients with first.”*
Physician 21

It was noted that WhatsApp was used


*“not only for dermatology but we use it across most departmental.”*
Physician 63 

With respect to the Health Professions Council of South Africa (HPCSA):


*“The role of telemedicine needs to be properly defined. HPCSA is out of touch.”*
Physician 77

### 4.5. Consent and Guidelines

Only 15 respondents (19%) were aware of the HPCSA General Ethical Guidelines for Good Practice in Telemedicine [35], 13 (9%) said they were aware of other guidelines (7 South African Medical Association, 1 Medical Defence Union, 5 Department of Health), and 9 (11%) were unaware of any guidelines. Nearly two-thirds of respondents (50, 62%) did not answer the question. Three respondents were aware of two or more guidelines. Of the total 81 responses, 2 did not obtain consent and 35 (43%) did not answer the consent question (Table 4).

### 4.6. Data Security

All but three respondents (91%) used one or more forms of security on their smartphones. This included the use of a password or PIN (66, 81%), fingerprint authentication (45, 55%), remote wiping (14, 17%), full device encryption (13, 16%), and a third-party lock screen app (3, 4%). Four people did not answer the question.

Thirty-three doctors (41%) became aware of IM for dermatology from a colleague at work, 19 (24%) from a dermatologist, 10 (12%) from a colleague at another hospital, and 1 from the KZ-N Department of Health. Sixteen (20%) thought of it themselves. Two did not answer. One doctor noted that the dermatologist guided doctors in the use of WhatsApp for dermatology. In deciding on their choice of a dermatologist, most chose to consult the Department of Dermatology at the medical school (34, 42%) or the dermatologist who conducted outreach at their site (14, 17%). Others sought the advice of a colleague (23, 28%), or chose to approach a dermatologist with whom they were acquainted (16, 20%). One (1%) had never sent messages to a dermatologist but to other medical officers, and one (1%) to a WhatsApp group with their dermatologist already part of the group. Some doctors (35, 43%) used more than one route for consults.

## 5. Discussion

The key findings of the study were that the informal use of WhatsApp for dermatology is widespread in district hospitals in KZ-N and that many doctors who used IM for teledermatology did not consider this to be telemedicine. It was used for consultation, second opinion, and advice from dermatologists in the KZ-N health service, the private sector, other provinces, and colleagues in their own and other hospitals. In the absence of any formal teledermatology service, doctors started using WhatsApp of their own volition, exploiting its ubiquity, low cost, and ease of use. They were satisfied with WhatsApp, felt that their patients were satisfied with the teledermatology provided, and expressed a desire for a formal service. Few doctors were aware of the HPCSA guidelines for telemedicine, and written consent was seldom obtained. Most doctors adopted data security measures on their phones. 

On the available data, dermatologists in the KZ-N Provincial Health Service responded to all of the unsolicited WhatsApp messages they received from doctors at district hospitals and mostly did so in under one hour. They typically received three short messages per case averaging 11 words each, with two to three images attached. The information and photographs provided were of sufficient quantity and quality for the dermatologist to make a diagnosis or request additional investigations. Additional photographs were required only once. All submitted cases received a response from the dermatologists in a median response time of 7.5 min which is in line with other studies—2.6 min [36], 4.9 min [37], 7 min [38], 7 min [39], and 52 min [40]. Over three quarters of the cases (252, 78%) received a response in under 30 min and 280 cases (87%) in under 60 min, favorably comparable to a previous study reporting 53% in under 60 min [40]. Nearly 80% of cases were managed solely through the chat messages and photographs without subsequent referral to the dermatologists or need for further investigation, similar to other studies—74.5% [37] and 61.8% [41]. Only 6% of cases required an additional face-to-face consultation with the dermatologist. 

No message contained any patient identifier and in only two cases, the patient’s face was obscured to avoid identification. Notification of consent having been obtained for the WhatsApp consultation was provided only once, similar to a finding reported by Meyer (3 of 109 cases) [42].

These findings are in line with the literature on the use of IM and more specifically WhatsApp in clinical services. Activities are seldom formally organized, and local pragmatic guidelines and rules evolve [16]. However, KZ-N doctors did indicate interest in a formalized service. How could the widespread acceptance of the informal use of WhatsApp for dermatology be transformed into an organized teledermatology service? A first step is to review the compliance of current practice in South Africa with the HPCSA’s General Ethical Guidelines for Good Practice in Telemedicine. 

Few referring doctors were aware of the HPCSA guidelines and their requirements, and few of those who did followed them. Of particular concern is the HPCSA’s onerous consent process that is counter to routine practice, and only mollified by frequent use of the instruction ‘should’. The study showed that consent, if obtained, was mostly verbal or implied but in all likelihood not sought, given that over 43% of respondents did not answer the question. This is in keeping with previous studies on consent practices of doctors in this province [43] and the literature on WhatsApp use in clinical practice [9]. Whether consent was recorded in the patients’ records is unknown, and in the absence of written consent, a ‘copy’ cannot have been given to the patient as HPCSA requires [35]. It has been suggested that the referring doctor should take a photograph of a consent form [33], if used, or the note placed in the patient’s record, and send it with the text message and other photographs.

The HPCSA guidelines indicate the consent process has to be: proper; informed; written, signed and witnessed; have the identity of the patient and both practitioners verified; and identify the location and practice number of the referring doctor and doctor consulted (or ‘consulting healthcare practitioner’ and ‘servicing healthcare practitioner, in the HPSCA terms) [35]. Each of these requirements pose difficulties in the routine practice of telemedicine (including IM), which by definition and practice is virtual. Further, ‘proper’ is not defined, ‘informed’ specifies requirements more akin to research than routine provision of care, ‘written, signed, and witnessed’ also go beyond expectations for routine provision of care, and ‘verification’ and ‘identification’ extend beyond everyday non-telemedicine referral. Furthermore, in the case of IM, the location and identity of an ‘on call’ doctor may not be known at the time of consent. In the study, both the sending and receiving doctors acted in good faith, probably in the belief that they were dealing with the appropriate person.

Further, the HPCSA’s consent process requires the referring doctor to explain to the patient the security measures taken regarding data transmission, its storage, and access— all issues about which the referring doctor may be unclear or even unaware [17]. In addition, consent should ideally be obtained in the patient’s mother tongue [43]. However, many indigenous African languages (of which there are at least 35 in South Africa; 11 being official languages) have not kept up with technology and do not have words for the relevant technical terms, and the validity of consent obtained in the absence of appropriate words in a language has been questioned [43].

The HPCSA Guidelines state that the clinician consulted must keep detailed records of the advice given and the information upon which it is based [35]. However, conflicting guidance is given regarding patient identification during electronic transmissions. HPCSA guidelines regarding confidentiality indicate policies and procedures should indicate which patient information be *included* in electronic communications (e.g., name, identification number, and type of transaction), while guidelines regarding security state personal identification should be *removed* when patient information is transmitted between sites [35]. The latter is in line with good IT governance practice and international practice when using WhatsApp [9]. However, how is the veracity of a patient record to be maintained without any form of patient identification in electronic transmissions to authenticate inclusion in a record? It has been suggested that chat messages stored on a phone constitute a medical record [9,18], but they are not part of patients’ records, nor are they secure as the phone may be lost or stolen, and may be deleted when the phone’s storage capacity is reached [18]. Furthermore, the doctor making the referral is also expected to keep a record of the electronic consultation in the patient’s record [35]. Yet how (in the still common absence of an electronic medical record) are the photographs to be entered into the notes other than by printing the images?

The application of artificial intelligence in the diagnosis of clinical images can be expected to become more widespread as technology advances and becomes routinely available in resource-constrained settings [44,45]. However, the infrastructural limitations in the current setting precluded their application. In the interim, and given the above ‘reality check’, the information obtained from this study provides crucial insight into the preliminary issues that need to be addressed when attempting to formalize an IM service within the public sector hospitals of KZ-N. Based upon the findings, it is proposed that initial steps include: (i) having an on-call dermatologist to serve all the district hospitals, or assigning a dermatologist to serve each hospital; (ii) keeping records of each case, which can initially be achieved by emailing the chats and photographs to a secure departmental email account; (iii) finding a simple solution to patient identification (in the absence of a unique identification system within the health service) since verifiable record keeping is not possible without an identifier; (iv) recording evidence of consent for the referral such as a photograph of the consent form or the patient’s notes documenting consent; and (v) developing guidelines and standard operating procedures for IM teledermatology in the public health sector of KZ-N.

## 6. Conclusions

The use of WhatsApp by clinicians for transmitting, storing, and sharing dermatological consults throughout KZ-N is extensive. Its use globally has increased during the COVID-19 pandemic. This informal use has resulted in substantial time and cost savings to both the physician and patient. However, greater formality (policy, procedures, and interim practical guidelines) are needed to ensure that the practice is conducted in an ethical manner. Further research is required to determine how best to formalize consults between district hospitals and dermatologists at the medical school, and to inform the Department of Dermatology about operational organization and staffing to meet the need. The described circumstances, issues, and consequences of the increasing use of IM for clinical applications, particularly in response to COVID-19, are likely to have been experienced throughout the globe, making the findings of this study of broad interest and value.

## Figures and Tables

**Table 1 ijerph-19-07462-t001:** Data extracted from WhatsApp chat referrals provided by dermatologists.

The Date and Time of the Data Sent by the Referring Doctor	Whether Patient Demographic Details Were Included in the Referral
The response time of the consultant.	Documentation of consent or similar.
The time of the final reply from the consultant.	The referring physician’s diagnosis and, if any, differential diagnosis.
The number of messages sent between referring doctor and consultant.	The consultant’s diagnosis and, if any, differential diagnosis.
The number of photographs sent.	The prescribed treatment by the consultant.
The quality of photographs.	If further investigations were requested.
Number of patient follow-up events per case.	If dermatologist requested patient transfer.

**Table 2 ijerph-19-07462-t002:** Summary of the content and response times of 322 cases with 1864 WhatsApp messages. (‘Case’ refers to all messages and photographs per completed referral).

		Time to First Response (MM:SS)	Time to Resolution (MM:SS)	Avg. No. Messages per Case (Range)	Avg. No. Words per Case (Range)	Avg. No. Photos per Case (Range)
Referring Dr				3.3 (0–33)	36 (0–70)	2.3 (0–9)
Dermatologist	median 25th quartile 75th quartile	7:51 2:34 24:10	16:00 5:47 55:25	2.6 (1–17)	35 (29–51)	-

**Table 3 ijerph-19-07462-t003:** Referral pathway.

Referral Pathway—Sent Consults to:	*n*	Always	Mostly	About Half the Time
A dermatologist in KZ-N Department of Health hospitals (medical school)	63	46	16	1
A dermatologist in other provinces	5	2	3	0
A dermatologist in private practice	2	2	0	0
Doctors at other public hospitals	28	15	12	1
Doctors at their own hospital	34	18	14	2

**Table 4 ijerph-19-07462-t004:** Form of consent obtained by doctors for instant messaging.

	Instant Messaging	
Form of Consent	Hpcsa Consent Requirements	
	Aware *n*	Unaware *n*
Written only	0	2
Written or verbal	1	1
Written, or verbal or implied	1	2
Verbal only	4	19
Verbal or implied	0	10
Implied only	0	4
None	1	1
No answer	8	27

## Data Availability

Restrictions apply to the availability of these data and are not publicly available due to Ethics Committee protocol and procedures.

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
