# Peer review of "An Audit and Survey of Informal Use of Instant Messaging for Dermatology in District Hospitals in KwaZulu-Natal, South Africa"

_ijerph, 2022, doi:10.3390/ijerph19127462_

Round 1
Reviewer 1 Report
Dear Authors, I have read your original paper very carefully and pleasantly: it is a cross-sectional study on how the use of messaging applications via social networks (Whatsapp and others) can be of help in situations and places without reference personnel. In this case of dermatologists. I found the article very original and inspiring. Here are just a few suggestions / comments to offer to the authors:
- I would advise the authors to add some original images (naturally without any personal information) in order to attract the reader more and make the reading of the paper more fluent.
- 2. I would advise the authors to make a brief mention of the situation not only dermatological, but (if any) also dermatopathological in the districts studied in this work. How are any skin biopsies of suspicious skin lesions managed? Who reads them? Is there a possibility of a similar system to this one?
- I suggest that the authors derive a paragraph in the discussion section in which a brief summary of potential applications of artificial intelligence to IM services is made from dermatological and dermatopathological images.
- Finally, I suggest to look for some bibliographical entries of the latest times on these topics of artificial intelligence and IM in dermatopathology, as well as dermatology. I'm sure the authors will make this last effort by making the manuscript even more captivating. In faith
Reviewer 2 Report
This article about “An Audit and Survey of Informal Use of Instant Messaging for Dermatology in District Hospitals in KwaZulu-Natal, South Africa” is attractive. The authors need to change or improve the next questions:
-Theoretical framework: it is too much short. It needs more citations, and more weight.
-Methods. It combines quantitative and qualitative tools. The authors can introduce a deeper analysis about qualitative tool. The “n” is small, but I understand it is impossible to increase it.
-Results: in general, they are right.
-Conclusions: The authors can expand it.
Round 2
Reviewer 1 Report
The authors responded very well to all requests. The article is ready for publication.